# Investigation of Pulsed Electric Field Conditions at Low Field Strength for the Tenderisation of Beef Topside

**DOI:** 10.3390/foods11182803

**Published:** 2022-09-11

**Authors:** Tomas Bolumar, Bo-Anne Rohlik, Janet Stark, Anita Sikes, Peter Watkins, Roman Buckow

**Affiliations:** 1CSIRO, 39 Kessels Road, Coopers Plains, Brisbane, QL 4108, Australia; 2Department of Safety and Quality of Meat, Max Rubner Institute, E. C. Baumann Strasse 20, 95326 Kulmbach, Germany; 3CSIRO, 671 Sneydes Road, Werribee, Melbourne, VIC 3030, Australia

**Keywords:** pulsed electric field, PEF, meat tenderisation, meat quality, novel food processing

## Abstract

Tenderness is the most critical eating quality trait of meat, and consequently, processing interventions for meat tenderisation have significant economic relevance. The objective of this study was to investigate pulsed electric field (PEF) conditions for the tenderisation of beef topside. The PEF settings included combinations of three field strengths (0.25, 0.50 and 1.00 kV/cm), two frequencies (20 and 100 Hz) and three treatment times (10, 30 and 50 ms). The effect of PEF on meat quality parameters (pH, drip loss, shear force, cook loss and colour) immediately after treatment and after storage (1 and 14 days at 4 °C) was evaluated. PEF did not affect meat tenderness after 1 day of chilled storage but resulted in a 5–10% reduction in the shear force in some cases (0.25–0.5 kV/cm) compared to the untreated control after 14 days of storage. Other quality traits (cook loss and colour) were not impaired. Thus, we concluded that PEF technology is a possible intervention to improve meat tenderness of beef topside after 2 weeks of storage.

## 1. Introduction

Tenderness is recognised as the most critical palatability trait for the eating satisfaction of meat [1], and consequently has a significant impact on its value and repeated purchase by consumers [2]. Colour is decisive in fresh meat on display, but once the meat is cooked becomes almost irrelevant, and the flavour is also of less importance compared to tenderness, provided that no off flavours are present. In this context, meat processors demand interventions that improve the tenderness of low-value muscles and ensure the consistency of high-value muscles [3,4,5]. Developing processing interventions for meat tenderisation is one of the keystones to boosting the profitability of the meat industry [6].

Tenderisation of meat using novel food processing technologies is a promising area to deliver alternative methods to the future of the meat industry [6,7]. As such, novel processing technologies, beyond heat treatments, have attracted considerable attention from the scientific community and the food industry in the last four decades. This has driven the development and scale-up of different food processing technologies, which are now available for testing in different fields and for specific applications, in particular related to ensuring food safety and quality [8,9].

PEF is an emerging processing technology based on the application of intermittent electric fields of low to high intensity (0.1–50 kV/cm) and short duration (from a few μs to several ms) [10] that could have potential for meat tenderisation. The application of PEF induces electroporation in biological materials, which forms reversible or irreversible pores in the cell membranes, as well as disruption of intracellular organelles and other structural changes which can result in the modification of food structure [10,11]. Furthermore, electroporation of the cell membrane can facilitate the mobility of intracellular constituents and thus enhance the contact between enzymes and substrates. Moreover, the catalytic activity of endogenous proteolytic enzymes in muscle structural proteins results in protein breakage and tenderness improvement of the meat [12].

Recently, different attempts to use PEF treatments to modify meat structure to improve meat tenderness have been performed. Some of these approaches have improved meat tenderness [13,14,15,16,17,18]. However, other studies have not observed any effect on meat tenderness after PEF application [14,18,19,20,21,22]. Hence, variable and contradictory results exist in the literature on the effect of PEF treatment for meat tenderisation. Therefore, the conditions for meat tenderisation by PEF are not clear and require fundamental research to optimise processing settings such as electric field strength, frequency and processing time [23,24].

Based on existing literature and our own unpublished data, it seems more likely that low field strengths (<1 kV/cm) in PEF processing are more effective than high field strengths [15,16,17]. Very recently, research has suggested that high-intensity PEF treatment (1.25 kV/cm) can negatively affect the quality of meat [25]. In addition, the frequency of PEF application could also play a role in the process of tenderisation. On the one hand, [13] and [15] observed that the higher the frequency (from 20 to 90 Hz), the higher the shear force reduction (i.e., improvement in tenderness) of beef topsides (*m. semimembranosus*), whereas the tenderness of m. longissimus lumborum from beef after PEF treatment was not frequency dependent. On the other hand, these same authors [16] reported that *m. longissimus lumborum* from beef became tougher with increasing frequency (from 20 to 90 Hz), whereas *m. semimembranosus* from beef was found to have up to a 21.6% reduction in shear force with PEF treatment. In addition to the field strength and pulse frequency, the treatment time is an equally important factor in PEF processing. Treatment time is related to pulse width and the number of pulses applied, and together with field strength determines the energy input into the system. Furthermore, treatment time includes processing time and energy consumption, with both having implications on the practicability of the treatment in an industrial environment (duration and cost).

The objective of the present work was to investigate the effect of PEF on the tenderisation of beef topside (*m. semimembranosus*). The treatment conditions comprised the testing of three low field strengths (0.25, 0.50 and 1.00 kV/cm) applied at two frequencies (20 and 100 Hz) for three different treatment times (10, 30 and 50 ms). The effect of the PEF treatment on selected meat quality parameters (pH, temperature, drip loss, Warner-Bratzler shear force, cook loss and colour) was evaluated immediately after PEF treatment and after 2 weeks of storage at 4 °C (at 1 and 14 days).

## 2. Materials and Methods

### 2.1. Meat Sampling and Preparation

Beef topside, *m. semimembranosus*, from both the left and right sides, was sourced from 6 domestic trade animals (mixed breed, 0–2 tooth, females) from a local butcher (Tasman Meats, Werribee, Victoria) at a post mortem time of 24–48 h. Carcass weight (235 ± 23 kg), pH (5.58 ± 0.03), temperature (5.56 ± 1.01) and topside weight (7.59 ± 0.44 kg) at harvest were recorded.

The muscle pH was measured twice in each muscle, and muscles with pH greater than 5.8 were rejected. The muscles were trimmed of excess fat and connective tissue and 19 portions, with an average weight of 58.1 ± 5.3 g and size of 65 × 30 × 25 mm, were cut from each muscle and randomly assigned to the different PEF treatments (Table 1) and two storage periods (1 and 14 days). One topside per animal was used per storage time, a total of n = 6 per storage period. Samples for PEF treatment were placed into low-density polyethylene bags with a layer thickness of 35 µm (Debro Chemicals, Brampton, ON, Canada) and stored in an ice slurry (0 °C) until required for PEF processing.

### 2.2. PEF Treatment

The PEF treatment of meat was performed using a Diversified Technologies Power Mod^TM^ 25 kW Pulsed Electric Field System (Diversified Technologies, Inc., Bedford, MA, USA), consisting of a PEF treatment chamber and a modulator cabinet [26]. The treatments were conducted in a treatment chamber designed and manufactured by the Commonwealth Scientific and Industrial Research Organisation (CSIRO) to allow the application of high electric field strengths to solid foods (Figure 1). All meat samples and the PEF treatment chamber were conditioned to approximately 1–2 °C in an ice slurry before PEF application. The PEF treatment chamber was dried of excess water and the samples were placed into the chamber with no air pockets at the surface of the electrodes and the PEF treatment was applied. The PEF treatment parameters are shown in Table 1. Square-wave pulses of 10 μs width at peak voltages of 1500, 3000 and 6000 V were applied, resulting in electrical field strengths of 0.25, 0.5 and 1.0 kV/cm, respectively (Table 1). The pulse repetition rate was set to either 20 or 100 Hz. PEF was applied to the meat for different treatment times resulting in several combinations of electric field and energy inputs (Table 1). The pulse shape, pulse width, voltage and electrical currents were recorded with an oscilloscope (#GDS-1102, GW Instek, Taipei, Taiwan), which was attached to the output port of the PEF system. These data were used to calculate the following values: energy per pulse (Wp), energy input and specific energy input according to the Equations (1)–(3), respectively.
Energy per pulse (Wp) = Current × Voltage × Pulse width(1)
Energy input = Wp × no. of pulses(2)
Specific energy input = Energy input / weight of the sample(3)

After PEF processing, all samples were put in polyethylene bags (Debro Chemicals, ON, Canada) and immersed in ice water for 5 min for cooling before immediate evaluation. Samples for storage tests were vacuum packaged in bags (VAC LS Pouch, Bemis, QL, Australia) and stored at 4 °C for either 1 or 14 days before evaluation upon opening.

### 2.3. Measurement of pH

The pH was measured by inserting a spearhead pH probe (IJ44C probe, Ionode, Pty Ltd., Tennyson, QLD, Australia) with a temperature probe (connected to a WP-80 pH-mV-Temperature meter, TPS Pty Ltd., Brendale, QLD, Australia) into the muscle. The pH meter was calibrated with pH 4.00 and pH 7.00 standards at 8 °C with pH measurements made per individual sample.

### 2.4. Colour Measurement

The colour of each sample block was measured on the cut end using a Minolta Colorimeter CR-300 (Minolta Co., Ltd., Japan). Measurements were taken before and after PEF treatment, after storage and prior to cooking as well as after cooking the meat. The instrument was equilibrated in a cool room (8 °C) and calibrated with a standard white plate under D65 illumination (Y = 92.0, x = 0.3163, y = 0.3328), observer angle 10°, aperture size 10 mm, prior to use. Colour measurements were taken directly on the muscle surface, on the sides of the cut pieces, avoiding areas of visible fat. Triplicate colour measurements were taken of each sample with the final data measured as *a** (redness), *b** (yellowness) and *L** (lightness). Three measurements were made on each sample.

### 2.5. Texture Measurement

The meat tenderness of the samples was measured with an Instron 5564 fitted with a 500 N load cell (Instron, Norwood, MA, USA) and a modified Warner-Bratzler shear device, as reported in [27,28]. The samples at each storage time point were cooked to an internal temperature of 80 °C for 30 min in a water bath. After cooking, the samples were cooled in an ice slurry for 20 min and then stored at 4 °C for 1 h prior to texture assessment. The samples were cut into a rectangular shape with dimensions of 15 mm width, 6.7 mm height, giving a cross-sectional area of 1.005 cm^2^, and at least 25 mm long to enable secure clamping of the sample into the holder. A triangular-shaped blade (thickness = 0.64 mm) attached to an overhead clamp was pulled up through the muscle fibres, perpendicular to the fibre direction, at a speed of 100 mm/min [27,28]. The maximum peak force, PF, was objectively determined by the Bluehill^®^ 3 software (Instron^®^, Illinois Tool Works Inc., Glenview, IL, USA), while the initial yield, IY, was determined by the operator as the height of the first peak of the curve. The difference between these measurements (PF-IY) was also calculated. Six measurements were conducted for each sample.

### 2.6. Drip Loss and Cook Loss

The weight of each sample was recorded before and after each PEF treatment to calculate the drip loss. The drip loss from each sample was also evaluated after storage (storage drip loss) by weight difference. After cooking for texture measurement (Section 2.5), samples were cooled in an ice slurry for 20 min and the weight recorded to determine cook loss.

### 2.7. Statistical Analysis

The data were analysed using a one-way analysis of variance (ANOVA) in R [29]. The first analysis evaluated the impact of PEF processing on several meat properties. Measurements of pH, weight and colour parameters (*L**, *a** and *b**) were compared before and immediately after the application of PEF treatment. An ANOVA was performed to discriminate the statistical differences for the time of measurement (i.e., before and after processing) and for the treatment applied (i.e., frequency, electric field strength, treatment time). The second analysis evaluated the impact of PEF treatment on meat quality parameters after storage. The measured parameters were peak force, initial yield, storage drip loss and cook loss, as well as the colour parameters, *L**, *a** and *b**. The ANOVA was performed to discriminate the statistical differences for the treatment applied (i.e., control, frequency, electric field strength, treatment time) and for the storage time (1 and 14 days) as factors.

## 3. Results

### 3.1. Immediate Effect of PEF Treatment on Meat Properties

The effect of PEF treatment on the meat properties immediately post treatment is presented in Table 2. The PEF treatment did not have a significant impact on pH (ΔpH, Table 2). Most of the pH values decreased very slightly in a range from −0.03 to −0.09 after treatment, though they were not significantly different (Table 2). The meat temperature increased proportionally to the energy input (Table 1), with a higher energy input resulting in a higher temperature in the meat (ΔT, Table 2). Most of the applied PEF treatments, apart from those of highest field strength (1 kV/cm) and longer treatment time (30 and 50 ms), resulted in a relatively low temperature increase (1–13 °C, see Table 2), which can equilibrate quickly to chilling temperatures (4–7 °C) once the product is placed in chilled storage. In addition, the longer treatment time resulted in a higher increase in temperature, which in turn increased the meat conductivity and so allowed a higher electric current to pass through the system, which overall increased the energy per pulse (Wp) (Table 1). Despite the increased energy input, all treatments resulted in an acceptable meat appearance except for the sample at the highest field strength (1 kV/cm) and longest treatment time (50 ms), which was partially burnt and so was not acceptable.

The drip loss (measured as Δweight) was slightly higher in PEF-treated meat, possibly because of the electroporation of the muscle cells, especially after PEF treatment at the highest field strength (1 kV/cm) and the longer treatment times of 30 and 50 ms (−0.7 and −0.9 g and −9.4 and −2.9 g, respectively, Table 2). In addition to electroporation, heat-induced protein denaturation could also result in increased drip loss in overheated samples. However, this effect on the drip loss was statistically significant only at a *p*-value < 0.1 for the variable treatment time (Table 1). For the rest of the PEF treatments, this additional drip loss was not higher than 0.1–0.2 g per sample compared to the control (−0.4 g), particularly at the low field strength (0.25–0.5 kV/cm, Table 2).

There was no immediate effect on the meat colour post PEF treatment (Table 3). Slight colour variations were observed, but these were within the expected variance for the beef used in the experiment: Δ*L** = 1.4 ± 0.7, Δ*a** = 0.3 ± 0.9 and Δ*b** = 1.2 ± 0.7 (Table 3). Therefore, the application of these PEF treatments is advantageous, as the raw meat colour was not changed nor impaired.

### 3.2. Effect of PEF Treatment on Meat Tenderness and Other Quality Parameters at 1 and 14 Days of Storage

The effect of the PEF treatments on meat tenderness at 1 day of storage is presented in Table 4. The peak force (PF) is the total amount of force required to shear through the sample and the initial yield (IY) is a measurement of the myofibrillar resistance. There was no significant improvement in tenderness (% PF reduction) after PEF treatment and 1 day of storage. In fact, some of the samples were tougher (−3.4 to −15.2%, Table 4). No effect of PEF treatment on meat tenderness as measured by shear force has also been reported [19], while meat toughening has been mentioned elsewhere [14,16].

The effect of PEF treatments on meat tenderness at 14 days of chilled storage is presented in Table 5. No improvement in tenderness for the control was found after 2 weeks of storage, whereas the PEF-treated samples corresponding to 0.25–0.5 kV/cm applied for 10 and 50 ms had an increase in the tenderness of 9.5, 8.5, 8.6 and 6.3% PF reduction, respectively. This would suggest that the PEF treatment induced tenderisation, most likely via proteolysis. In agreement with this, [15,16] also found tenderisation of beef topside with PEF treatment, with a reduction in the peak shear force reaching 20%. In addition, an interaction of tenderness was found to exist with the frequency (20 or 100 Hz) (*p* < 0.001). A low frequency (20 Hz) affected the meat tenderness after 1 day. The % PF reductions in the treatments at 0.25 kV/cm and 0.5 kV/cm for 50 and 30 ms applied at 20 Hz were positive (6.9, 7.3, 6.8 and 1.9% PF reduction, respectively) in comparison to the application of the same PEF conditions but at 100 Hz, which were, in contrast, negative (−4.2, −11.0, −14.3 and −13.3% respectively) (Table 4). After 14 days of storage, it was not conclusive whether frequency (20 or 100 Hz) affected meat tenderness (Table 5).

There were no major changes in pH, cook loss and colour (*L**, *a** and *b**) after PEF treatment and 1 day of storage compared to the untreated control (Table 6). There were also no significant differences for the measured colour (*L**, *a** and *b**) of cooked meat between the treated and control samples at 1 and 14 days. The storage drip loss trended slightly higher in the PEF-treated samples compared to the control (plus 0.1–1.3%) with an average of plus 0.6%, although these differences were not statistically significant (Table 6). Thus, there was no significant difference in the storage drip loss in the PEF-treated samples compared to the control (Table 6).

There were no major changes in pH, cook loss and colour (*L**, *a** and *b**) of the meat after PEF treatment and chilled storage for 14 days (Table 7). Again, a trend was apparent, although not statistically significant, that the storage drip loss for the treated samples was slightly higher than the control. This additional storage drip loss after 14 days of storage compared to the control (7.2 ± 0.5%) was in the range of an additional 0.3–3.2% and an average of plus 0.6% (Table 7). The cooking loss of the control (32.6 ± 0.8%) was in the same range as PEF-treated samples (30.9–32.5% with an average of 31.8%) (Table 7).

## 4. Discussion

The application of PEF treatment for meat tenderisation is a relatively new topic in food processing. The first detailed study on the use of PEF for improving the tenderness of fresh beef (*m. semitendinosus*) was recently published [22]. These authors reported no effect of PEF treatment (electric field strength, 1.1–2.8 kV/cm; frequency, 5–200 Hz; pulse width, 20 µs; and pulse number, 152–300) on meat tenderness but a significant weight loss was found as a result of additional drip loss. More recent studies by Arroyo et al., 2015 [19] (1.4 kV/cm, 10 Hz, pulse width 20 μs, 300 and 600 pulses), Faridnia et al., 2014 [20] (0.2–0.6 kV/cm, 1–50 Hz, 20 µs), and Faridnia et al., 2015 [21] (1.4 kV/cm, 50 Hz, pulse width 20 μs, 1032 pulses), showed similar results with no effect on tenderness after PEF treatment and a trend for higher drip loss. In contrast, other authors have reported improvements in meat tenderness: Bekhit et al., 2014 [13] (0.27–0.57 kV/cm, 20, 50 and 90 Hz, pulse width 20 μs, 607,1529 and 2726 pulses), Bekhit et al. (2016) [14] (0.76 kV/cm, 90 Hz, pulse width 20 μs, 2700 pulses), Suwandy et al., 2015 [15] (0.38–0.77 kV/cm, 20, 50 and 90 Hz, pulse width 20 μs, 607,1529 and 2726 pulses), Suwandy et al., 2015 [16] (0.25–0.6 kV/cm, 20–90 Hz, pulse width 20 μs, 607–2726 pulses), Suwandy et al., 2015 [17] (0.59–0.73 kV/cm, 90 Hz, pulse width 20 μs, number of pulses is unknown as it was not specified in the paper), and Suwandy et al., 2015 [18] (0.76 kV/cm, 90 Hz, pulse width 20 μs, number of pulses is unknown as it was not specified).

Based on these previous studies, our work focused on assessing the following PEF parameters (0.25–1.0 kV/cm, 20 and 100 Hz, pulse width 10 μs and 1000, 3000 and 5000 pulses) and provides valuable information in relation to the optimal PEF settings for meat tenderisation. In relation to field strength, it was confirmed that low values in the range of 0.25 to 0.50 kV/cm were more effective for meat tenderisation. This agrees with previously reported work where low field strengths have also been employed (0.27–0.77 kV/cm) [13,14,15,16,17,18]. In contrast, when the field strength applied was higher than 1 kV/cm (e.g., 1.1–2.8 kV/cm, 1.4 kV/cm and 1.4 kV/cm, respectively, from [19,21,22]), no improvements in meat tenderness were observed, or even negative effects were observed on meat quality [25]. Moreover, it has been reported that the particular frequency of the electric pulse in some cases affected the meat texture: [13] and [15] observed that the higher the frequency (from 20 to 90 Hz), the higher the shear reduction in *m. semimembranosus* (i.e., more improvement in tenderness), whereas after PEF treatment, the tenderness of *m. longissimus lumborum* from beef was not frequency-dependent. In contrast, the same authors described in another study that *m. longissimus lumborum* became tougher with increasing frequency (from 20 to 90 Hz) [16].

This present work also showed that there was a statistical correlation between tenderness (peak force and initial yield) and the applied frequency. However, given the percentage of PF reduction (Table 5), it was not conclusive which applied frequency improved tenderness the most. Moreover, the treatment time was tested in the range of 10–50 ms, and again considering the percentage of PF reduction (Table 5), it was also not clear which treatment time was more effective. Suwandy et al., 2015 [15,16], also reported there was no interaction of shear force with treatment time and intensity. Hence, shorter treatment times would be preferred for the application of PEF in the industry, as they require shorter processing times and lower energy consumption. Shorter treatment times can be achieved by using high frequencies and fewer pulses. Overall, the recommendation for future work would be to further continue the optimisation of a low-energy PEF treatment at 0.25 kV/cm for 10–30 ms, preferably applied at 100 Hz.

Nevertheless, there are still inconsistencies within the literature regarding the application of PEF for meat tenderisation. The use of relatively low field strengths (<1.0 kV/cm) seems to be consistent, though more research is still needed to better understand the impact of particular PEF settings on meat structure. The effect of frequency and treatment time need to be determined more accurately, although different meat and muscle types may require different experimental PEF parameters. The intrinsic variability of meat may well contribute to the need for different settings; for example, higher collagen is present in topside (*m. semimembranosus*), which will impact its muscle structure compared to loin (*m. longissimus*) [30,31]. As meat is a highly variable and complex food matrix, a myriad of other factors such as age, feed, physical activity, collagen amount and degree of cross-linking, fat amount and muscle proteolytic degradation will contribute to the structural integrity of different muscle types [32]. As a result, different muscle cuts from different species could require different PEF treatment settings. This indeed has been reported by other researchers. Suwandy et al., 2015 [18], observed that PEF treatment (0.76 kV/cm, 90 Hz, 20 µs) had a positive effect on tenderisation of beef *m. longissimus lumborum* (2.5 N reduction in the shear force with every extra application of PEF treatment), while *m. semimembranosu**s,* in the same study, was not affected by any PEF treatment. In contrast, Bekhit et al., 2016 [14], reported a more beneficial effect of PEF treatment on the tenderness in hot-boned *m. semimembranosus* muscles than in *m. longissimus lumborum* muscles.

Another meat quality trait of practical relevance is the water retention expressed as drip loss. In this present study, drip loss of PEF-treated meat was not significantly increased, yet a trend for higher losses than in untreated samples was observed, which is in agreement with several other studies [14,16,18,21,22]. This increased drip loss was expected due to the electroporation caused by PEF treatments, whereas cooking loss seems consistently not to be affected by PEF treatment [13,16,17,19,21]. Water retention in meat is an important determinant of meat-eating quality (for instance, affecting meat juiciness), value (weight loss) and appearance, and for that reason, it must be carefully assessed when evaluating the application of PEF for meat tenderisation in future trials. Different approaches in meat processing and the addition of salts might counteract the excess drip loss, although the product’s classification will transform from being a ‘fresh meat’ to becoming a ‘further-processed meat product’ after the application of some of these interventions (e.g., immersion/tumbling with marinades containing phosphates). Although these interventions would be associated with a lower price for the meat, the PEF treatment combined with a marination step (e.g., using a brine solution) could address the possible PEF-induced drip loss and ensure enhanced tenderness and juiciness in ready-to-eat meat products.

Regarding the mechanisms behind the tenderisation, the relatively high-intensity electrical current applied in PEF treatment could cause physically driven contractions within the muscle, causing damage to the myofibrillar structure. PEF treatment at low field strength (less than 2 kV/cm and 20–40 pulses) has been shown to have a considerable effect on the microstructure of muscles from salmon and chicken, i.e., the muscle cells decreased in size and gaping occurred [33]. Other authors have also described alteration of the muscle microstructure. O’Dowd et al., 2013 [22], reported structural changes as indicated by fragmentation of myofibrils and observed in scanning electron microscopy, while Faridnia et al., 2014 and 2015 [20,21], described that when PEF was used alone or with freezing as a pre-treatment step, this led to the development of more porous beef muscles. These structural changes are noteworthy as this may account for the tenderisation effect that was found when meat was allowed to age for a reasonable period of time (2–3 weeks) after the PEF treatment, which was not the case in the work of O’Dowd et al., 2013, and Faridnia et al., 2014 [20,22]. If true, even though there were changes observed in the microstructure, this would explain why there was no meat tenderisation found by these researchers [20,22]. The authors of [15,16] observed a reduction in shear force with PEF-treated meat, but the treated meat was already more tender at day 3, indicating that ageing was not required for improving the meat tenderness in that study. In addition, PEF could promote the release of Ca^2+^ and favour early post mortem μ-calpain activity, in addition to the possibility of release of cathepsins from lysosomes. An enhancement of proteolysis maximises the tenderisation of meat over chilled storage (7–21 days or even longer). Nevertheless, the proteolytic mechanisms after PEF treatment of meat are not well understood, although some work has studied the proteolytic pattern of beef muscle proteins resulting from PEF treatment [14,15,17,18]. There seems to be an enhancement of muscle proteolysis, as shown by increased degradation of troponin-T and desmin, which is typical for the natural ageing of meat. Both groups of researchers suggested that the accelerated proteolysis of meat might contribute to the reduction in WB shear force in meat after PEF treatment. This is in contrast with another study by one of these groups where they have also hypothesised that due to a decrease in proteolysis with extra application of PEF treatments, excessive intensity of PEF treatment could be detrimental to proper meat ageing due to the inactivation of proteolytic endogenous muscle enzymes [14].

The use of PEF treatments at low field strengths (~0.25–0.50 kV/cm), rather than high field strengths, is advantageous per se. From the technological viewpoint, applying PEF treatments at low field strengths is technically less challenging and will also have a lower energy demand due to the use of lower voltages and electric currents. The use of lower field strengths also mitigates the need for highly conductive materials with higher operational temperatures as well as the associated safety measures. This makes it easier to scale up the process in an industrial context, with an associated reduction in energy consumption in comparison to the use of higher field strength PEF treatments. Although this work has demonstrated that PEF can potentially be used as an intervention to improve meat tenderness after 2 weeks of storage at 4 °C, much further work is still needed to understand the molecular mechanisms of PEF-induced muscle structure modifications and endogenous proteolysis in different muscle cuts and species. Additionally, the elaboration of precise cost models is a prerequisite for the decision-making process by the meat industry before investment in PEF technology. This work will aid in understanding the impact that PEF treatment has on meat tenderisation, and thus will assist in the development of customised PEF treatment settings required for the tenderisation of specific meat cuts and allow for the full exploitation of this technology for meat tenderisation by the industry.

## 5. Conclusions

The application of PEF treatments (0.25–0.5 kV/cm, 20–100 Hz and 1000–5000 pulses) has shown some improvement in tenderness (5–10% shear force reduction) of beef topside (*m. semimembranosus*) in some cases after 14 days cold storage. PEF treatment may have the potential for tenderisation under the correct processing settings, which still need to be determined. This slight improvement in meat tenderness was observed after applying relatively low electrical field strengths (0.25–0.5 kV/cm) rather than using an electrical field strength of 1 kV/cm or higher. Application of 0.25 kV/cm for short treatment times (10 ms or 1000 pulses) resulted in a significant improvement in tenderness, whereas at 0.5 kV/cm, longer treatment times (50 ms or 5000 pulses) were needed to achieve significant tenderisation after 14 days. The tenderisation could occur due to increased proteolytic activity and/or structural modification of the meat, which becomes evident during storage. The PEF treatment did not significantly impair other important meat quality traits (e.g., colour, storage drip loss and cooking loss). Further research is required to validate the optimal PEF treatment settings on different muscle types of economic importance and to understand the impact PEF has at a biochemical, molecular and structural level on meat.

## Figures and Tables

**Figure 1 foods-11-02803-f001:**
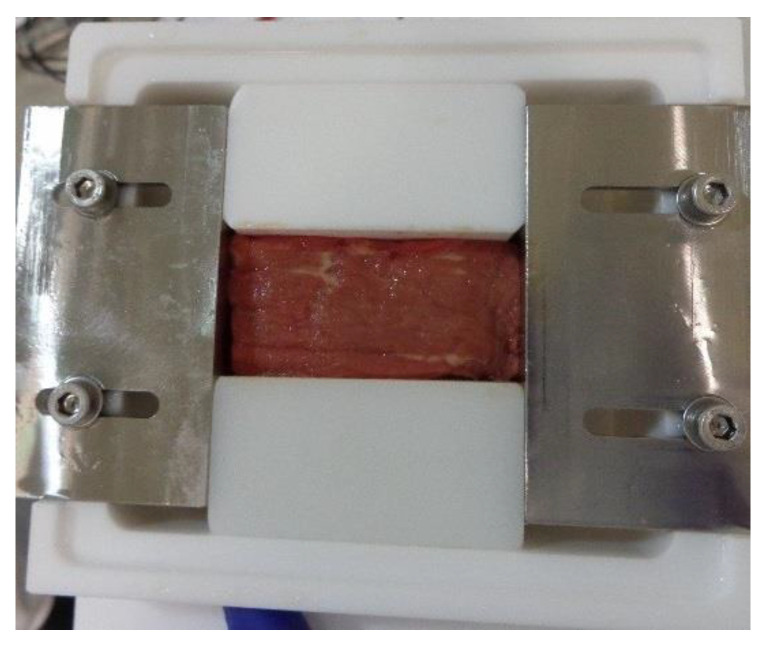
CSIRO’s PEF treatment chamber for treatment of solid foods.

**Table 1 foods-11-02803-t001:** Pulsed electric field (PEF) treatment settings.

Treatment Conditions									
Field Strength	Frequency	Treatment Time	Voltage ^#^	Current ^#^	Pulse Width ^#^	Energy Pulse ^##^	No Pulses	Energy Input ^##^	Specific Energy Input ^##^
(kV/cm)	(Hz)	(ms)	(V)	(A)	(µs)	(J/pulse)		(J)	(J/g)
0.25	100	50	1504	9.3	10	0.14	5000	698	12.0
0.25	20	50	1488	9.3	10	0.14	5000	690	11.9
0.25	100	30	1488	9.4	10	0.14	3000	421	7.2
0.25	20	30	1488	11.4	10	0.17	3000	507	8.7
0.25	100	10	1504	8.3	10	0.13	1000	125	2.2
0.25	20	10	1488	9.4	10	0.14	1000	141	2.4
0.50	100	50	3136	21.1	10	0.66	5000	3312	57.0
0.50	20	50	3104	22.7	10	0.71	5000	3526	60.6
0.50	100	30	3104	22.1	10	0.69	3000	2056	35.4
0.50	20	30	3168	16.3	10	0.52	3000	1551	26.7
0.50	100	10	3136	18.2	10	0.57	1000	572	9.8
0.50	20	10	3136	17.3	10	0.54	1000	542	9.3
1.00	100	50	6160	56.8	10	3.50	5000	17494	300.9
1.00	20	50	6160	54.6	10	3.36	5000	16817	289.2
1.00	100	30	6240	52.6	10	3.28	3000	9847	169.4
1.00	20	30	6240	51.6	10	3.22	3000	9660	166.1
1.00	100	10	6320	47.2	10	2.98	1000	2983	51.3
1.00	20	10	6400	44.0	10	2.82	1000	2816	48.4
Control (untreated)			-	-	-	-	-	-	-

^#^: Voltage, current and pulse width were measured during PEF processing. ^##^: Energy per pulse, energy input and specific energy input were calculated according to Equations (1)–(3), respectively, from Materials and Methods Section.

**Table 2 foods-11-02803-t002:** Immediate effect of PEF treatment on meat pH, temperature and drip loss from beef topside (x (mean) ± SE (standard error), n = 6).

Treatment Conditions			Meat Properties ^#^					
Field Strength	Frequency	Treatment Time	ΔpH		ΔT		ΔWeight ^##^	
(kV/cm)	(Hz)	(ms)			(°C)		(g)	
			x	SE	x	SE	x	SE
0.25	100	50	−0.05	0.02	3.6	0.3	−0.5	0.2
0.25	20	50	−0.03	0.04	3.7	0.4	−0.4	0.1
0.25	100	30	−0.06	0.01	2.3	0.2	−0.5	0.1
0.25	20	30	−0.03	0.03	2.7	0.2	−0.4	0.1
0.25	100	10	−0.04	0.02	1.8	0.4	−0.4	0.1
0.25	20	10	−0.03	0.02	1.7	0.3	−0.2	0.2
0.50	100	50	−0.08	0.01	13.3	0.6	−0.3	0.1
0.50	20	50	−0.08	0.02	13.8	0.6	−0.5	0.1
0.50	100	30	−0.07	0.02	8.4	0.3	−0.3	0.1
0.50	20	30	−0.06	0.01	8.0	0.3	−0.4	0.1
0.50	100	10	−0.06	0.02	3.9	0.4	−0.3	0.1
0.50	20	10	−0.05	0.02	2.9	0.4	−0.4	0.1
1.00	100	50	0.13 ^###^	-	49.5	-	−2.9	-
1.00	20	50	0.26 ^###^	-	67.1	-	−9.4	-
1.00	100	30	0.02	0.02	39.5	0.9	−0.9	0.1
1.00	20	30	0.00	0.02	36.4	1.5	−0.7	0.1
1.00	100	10	−0.06	0.02	12.5	0.5	−0.4	0.2
1.00	20	10	−0.09	0.02	11.5	0.5	−0.4	1.3
Control (untreated)			--	--	--	--	−0.4	0.1
Statistical significance ^####^								
Field strength			***		***		-	
Frequency			-		-		-	
Treatment time			-		***		+	

^#^: Meat properties, expressed as Δ, are the difference between the value measured after and before treatment; ^##^: Δweight is the measure of weight difference (i.e., weight loss or drip loss). The more negative the value for Δweight, the higher the drip loss due to the treatment itself. ^###^: These treatments led to a partially burnt product which was not acceptable; ^####^: n.s. (non-significant); + *p* < 0.1; *** *p* < 0.001.

**Table 3 foods-11-02803-t003:** Immediate effect of PEF treatment on meat colour (*L**, *a**, *b**) from beef topside (x (mean) ± SE (standard error), n = 6).

Processing Conditions			Colour ^#^					
Field Strength	Frequency	Treatment Time	Δ*L** (after–before)		Δ*a** (after–before)		Δ*b** (after–before)	
(kV/cm)	(Hz)	(ms)						
			x	SE	x	SE	x	SE
0.25	100	50	1.9	0.8	−0.4	0.8	0.4	0.4
0.25	20	50	1.7	1.0	0.6	1.0	1.1	0.6
0.25	100	30	1.7	0.8	−0.4	0.5	0.4	0.3
0.25	20	30	1.4	0.4	0.1	0.3	0.3	0.2
0.25	100	10	1.7	0.7	−0.1	0.8	0.6	0.6
0.25	20	10	1.2	0.8	0.4	0.5	0.3	0.5
0.50	100	50	1.6	0.4	−0.2	0.5	1.2	0.6
0.50	20	50	0.9	0.6	−0.4	0.5	0.9	0.6
0.50	100	30	2.0	0.5	−0.3	0.8	1.1	0.4
0.50	20	30	1.3	0.9	0.3	0.7	0.9	0.4
0.50	100	10	2.3	1.3	−0.4	0.8	1.0	0.4
0.50	20	10	1.5	0.3	0.2	0.4	1.0	0.5
1.00	100	50	n.d. ^##^	-	n.d.	-	n.d.	-
1.00	20	50	n.d.	-	n.d.	-	n.d.	-
1.00	100	30	2.5	1.1	−0.6	1.1	1.4	0.6
1.00	20	30	1.2	0.9	−2.5	0.7	−0.2	0.4
1.00	100	10	0.3	3.7	−1.9	0.7	0.7	0.5
1.00	20	10	2.5	1.2	−0.7	1.1	1.3	0.5
Control (untreated) ^###^			1.4	0.7	0.3	0.9	1.2	0.7
Statistical significance ^####^								
Field strength			-		***		+	
Frequency			-		-		-	
Treatment time			-		-		-	

^#^: Colour parameters expressed as Δ are the difference between the value measured after and before treatment. ^##^: These treatments led to a partially burnt product which was not acceptable. ^###:^ The control samples, even though they were not treated, were also measured twice as per treated samples. This represents the natural variation of meat colour without applying any treatment and colour changes below this value are within the normal variation of meat; ^####:^ n.s. (non-significant); + *p* < 0.1; ****p* < 0.001; n.d.: not determined.

**Table 4 foods-11-02803-t004:** Effect of PEF treatment on meat tenderness from beef topside after 1 day of storage (x (mean) ± SE (standard error), n = 6).

Processing Conditions			Texture Measurements							
Field Strength	Frequency	Treatment Time	Peak Force (PF)		Initial Yield (IY)		PF-IY		% PF Reduction ^#^	
(kV/cm)	(Hz)	(ms)	(N)		(N)		(N)		[%]	
			x	SE	x	SE	x	SE	x	SE
0.25	100	50	51.2	4.6	43.7	4.7	7.5	2.6	−4.2	0.4
0.25	20	50	45.7	4.4	35.7	5.3	10.0	3.6	6.9	0.7
0.25	100	30	54.5	5.6	49.6	5.5	4.9	2.9	−11.0	1.1
0.25	20	30	45.5	5.7	35.6	5.6	9.9	2.7	7.3	0.9
0.25	100	10	50.6	4.8	45.6	5.1	5.0	2.2	−3.1	0.3
0.25	20	10	50.7	5.0	41.6	6.0	9.1	2.5	−3.2	0.3
0.50	100	50	56.1	4.9	43.7	6.3	12.4	3.3	−14.3	1.3
0.50	20	50	45.7	4.6	36.4	5.1	9.4	2.4	6.8	0.7
0.50	100	30	55.6	5.1	47.9	4.7	7.7	2.9	−13.3	1.2
0.50	20	30	48.2	5.5	38.3	7.0	9.9	2.8	1.9	0.2
0.50	100	10	52.9	4.3	43.8	4.9	9.1	3.1	−7.7	0.6
0.50	20	10	50.7	4.8	37.2	4.8	13.5	2.9	−3.4	0.3
1.00	100	50	n.d. ^##^	-	n.d.	-	n.d.	-	n.d.	-
1.00	20	50	n.d. ^##^	-	n.d.	-	n.d.	-	n.d.	-
1.00	100	30	55.5	5.2	48.9	5.6	6.5	2.3	−13.0	1.2
1.00	20	30	50.9	5.5	41.2	5.7	9.7	2.9	−3.7	0.4
1.00	100	10	51.6	5.6	41.7	6.2	9.9	2.5	−5.0	0.6
1.00	20	10	56.5	3.8	47.8	5.1	8.7	2.9	−15.2	1.0
Control (untreated)			49.1	3.8	41.3	4.8	7.8	2.1	0.0	0.0
Statistical significance ^###^										
Field strength			***		+		***			
Frequency			***		***		***			
Treatment time			-		-		*			

^#^: Peak force reduction—a positive value means improvement in tenderness and a negative value means a reduction in tenderness in comparison to the untreated control. ^##^: These treatments led to a partially burnt product which was not acceptable; ^###^: n.s. (non-significant); + *p* < 0.1; * *p* < 0.05; 0.01; ****p* < 0.001; n.d.: not determined.

**Table 5 foods-11-02803-t005:** Effect of PEF treatment on meat tenderness from beef topside after 14 days of storage (x (mean) ± SE (standard error), n = 6).

Processing Conditions			Texture Measurements							
Field Strength	Frequency	Treatment Time	Peak Force (PF)		Initial Yield (IY)		PF-IY		% PF Reduction ^#^	
(kV/cm)	(Hz)	(ms)	(N)		(N)		(N)		[%]	
			x	SE	x	SE	x	SE	x	SE
0.25	100	50	45.7	4.5	32.4	4.2	13.3	4.0	9.6	0.9
0.25	20	50	51.1	4.1	39.6	3.5	11.5	3.2	−1.1	−0.1
0.25	100	30	53.3	2.4	41.1	3.5	12.2	3.2	−5.4	−0.2
0.25	20	30	47.7	2.8	33.9	3.6	13.7	3.6	5.8	0.3
0.25	100	10	46.3	3.1	34.3	2.7	11.9	3.4	8.5	0.6
0.25	20	10	45.8	3.5	32.1	3.7	13.7	2.4	9.5	0.7
0.50	100	50	46.2	4.6	33.7	4.8	12.6	2.8	8.6	0.9
0.50	20	50	47.4	4.2	36.1	5.1	11.2	3.2	6.3	0.6
0.50	100	30	50.5	3.2	39.0	4.4	11.4	3.0	0.2	0.0
0.50	20	30	48.8	3.9	34.4	3.7	14.4	4.0	3.5	0.3
0.50	100	10	48.9	5.3	35.1	4.8	13.9	3.4	3.2	0.4
0.50	20	10	50.9	3.0	38.8	3.7	12.1	3.0	−0.6	0.0
1.00	100	50	n.d. ^##^	-	n.d.	-	n.d.	-	n.d.	-
1.00	20	50	n.d.	-	n.d.	-	n.d.	-	n.d.	-
1.00	100	30	50.8	3.1	35.6	3.5	15.2	4.0	−0.4	0.0
1.00	20	30	47.6	4.2	33.9	4.9	13.7	3.2	5.9	0.5
1.00	100	10	47.4	3.8	36.6	5.0	10.8	2.7	6.3	0.5
1.00	20	10	50.3	6.3	36.4	5.5	13.9	2.8	0.6	0.1
Control (untreated)			50.6	3.9	35.8	4.2	14.8	3.5	0.0	0.0
Statistical significance ^###^										
Field strength			-		*		-			
Frequency			-		-		-			
Treatment time			*		-		*			

^#^: Peak force reduction—a positive value means improvement in tenderness and a negative value means a reduction in tenderness in comparison to the untreated control. ^##^: These treatments led to a partially burnt product which was not acceptable; ^###^: n.s. (non-significant); + *p* < 0.1; * *p* < 0.05; n.d.: not determined.

**Table 6 foods-11-02803-t006:** Effect of PEF treatment on different meat quality parameters (pH, storage drip loss, cook loss and colour (*L**, *a** and *b**)) from beef topside after 1 day of storage (x (mean) ± SE (standard error), n = 6).

Processing Conditions			Quality Parameters											
Field Strength	Frequency	Treatment Time	pH		Storage Drip Loss		Cook Loss		*L**		*a**		*b**	
(kV/cm)	(Hz)	(ms)			(%)		(%)							
			x	SE	x	SE	x	SE	x	SE	x	SE	x	SE
0.25	100	50	5.48	0.03	4.0	0.6	29.2	1.0	35.5	1.2	18.6	0.4	4.4	0.7
0.25	20	50	5.46	0.02	3.6	3.1	30.9	2.7	34.1	0.6	18.7	0.4	3.6	0.4
0.25	100	30	5.50	0.03	3.6	0.5	29.7	1.7	36.7	0.9	20.1	0.3	5.9	0.6
0.25	20	30	5.44	0.01	3.6	0.9	27.6	1.4	34.5	1.2	18.7	0.4	4.0	0.7
0.25	100	10	5.47	0.01	3.7	0.5	31.8	1.4	35.1	0.8	18.7	0.5	4.4	0.9
0.25	20	10	5.47	0.02	5.0	2.8	29.8	1.1	35.7	1.1	19.3	0.5	4.9	0.3
0.50	100	50	5.51	0.04	3.1	0.4	31.1	2.0	36.3	1.3	17.5	1.0	4.5	1.0
0.50	20	50	5.49	0.03	3.1	0.4	28.2	2.2	35.7	1.1	18.8	0.9	4.4	1.0
0.50	100	30	5.52	0.02	2.7	0.4	29.5	1.4	35.8	0.8	17.9	1.1	4.8	0.5
0.50	20	30	5.45	0.02	3.6	0.5	29.4	1.3	36.4	0.8	19.1	0.5	4.9	0.8
0.50	100	10	5.50	0.03	3.5	0.5	30.3	1.3	36.2	0.8	18.2	0.8	5.0	0.6
0.50	20	10	5.48	0.02	2.8	0.5	30.6	1.6	30.7	3.7	18.9	0.8	5.0	1.0
1.00	100	50	n.d. ^#^	-	n.d.	-	n.d.	-	n.d.	-	n.d.	-	n.d.	-
1.00	20	50	n.d.	-	n.d.	-	n.d.	-	n.d.	-	n.d.	-	n.d.	-
1.00	100	30	5.53	0.03	4.9	0.5	29.9	1.1	36.5	0.7	20.3	0.9	5.6	0.3
1.00	20	30	5.53	0.01	5.0	0.3	30.8	1.3	37.2	1.2	18.4	1.2	5.4	0.8
1.00	100	10	5.48	0.02	3.2	0.3	30.2	2.6	36.5	1.1	18.1	0.4	4.7	0.8
1.00	20	10	5.51	0.03	3.0	0.3	30.6	1.7	37.2	0.8	18.7	0.8	5.4	0.9
Control (untreated)			5.49	0.02	2.7	0.3	29.1	1.2	29.1	1.2	17.7	1.0	6.6	0.5
Statistical significance ^##^														
Field strength			-		-		-		**		-		-	
Frequency			-		-		-		-		-		-	
Treatment time			-		-		-		-		+		-	

^#^: These treatments led to a partially burnt product which was not acceptable; ^##^: n.s. (non-significant); + *p* < 0.1; ** *p* < 0.01; n.d.: not determined.

**Table 7 foods-11-02803-t007:** Effect of PEF treatment on different meat quality parameters (pH, storage drip loss, cook loss and colour (*L**, *a** and *b**)) from beef topside after 14 days of storage (x (mean) ± SE (standard error), n = 6).

Processing Conditions			Quality Parameters											
Field Strength	Frequency	Treatment Time	pH		Storage Drip Loss		Cook Loss		*L**		*a**		*b**	
(kV/cm)	(Hz)	(ms)			(%)		(%)							
			x	SE	x	SE	x	SE	x	SE	x	SE	x	SE
0.25	100	50	5.34	0.02	7.6	0.7	31.5	1.3	36.7	0.2	17.2	1.4	6.7	0.5
0.25	20	50	5.36	0.02	8.1	0.5	31.3	1.1	37.9	1.1	16.2	0.9	6.6	0.7
0.25	100	30	5.34	0.03	8.1	0.7	32.2	1.1	38.7	1.0	15.7	1.2	7.2	0.7
0.25	20	30	5.31	0.04	7.9	0.5	32.1	1.6	36.1	0.9	17.3	0.9	5.5	0.9
0.25	100	10	5.36	0.02	8.0	0.9	32.5	0.9	38.7	1.2	16.5	1.2	6.6	1.0
0.25	20	10	5.32	0.04	8.1	0.5	32.3	1.0	36.4	0.6	17.4	0.8	5.9	0.8
0.50	100	50	5.37	0.02	8.2	1.0	31.1	1.1	38.0	0.8	17.2	1.1	6.5	0.5
0.50	20	50	5.30	0.03	7.6	0.5	31.9	1.4	37.9	1.2	18.4	1.3	6.7	0.6
0.50	100	30	5.40	0.02	7.5	0.9	31.7	1.7	37.3	0.3	18.3	0.4	6.1	0.2
0.50	20	30	5.35	0.03	6.9	0.4	31.8	1.9	37.4	1.0	17.1	1.2	6.5	0.7
0.50	100	10	5.35	0.02	8.1	0.9	32.2	1.2	37.3	0.8	17.3	0.9	6.1	0.6
0.50	20	10	5.33	0.02	8.7	0.7	31.5	0.6	38.6	1.0	15.5	0.9	7.1	1.1
1.00	100	50	n.d.^#^	-	n.d.	-	n.d.	-	n.d.	-	n.d.	-	n.d.	-
1.00	20	50	n.d.	-	n.d.	-	n.d.	-	n.d.	-	n.d.	-	n.d.	-
1.00	100	30	5.33	0.02	10.5	1.0	30.9	1.0	36.6	0.5	17.5	0.8	5.6	0.7
1.00	20	30	5.35	0.03	8.4	0.3	31.8	1.3	37.2	0.7	16.6	1.0	6.4	0.6
1.00	100	10	5.35	0.03	7.2	0.9	31.7	1.6	37.1	1.1	17.0	1.4	6.3	0.9
1.00	20	10	5.34	0.01	8.1	0.8	31.6	1.5	38.2	1.0	15.3	1.2	6.7	1.0
Control (untreated)			5.36	0.01	7.2	0.5	32.6	0.8	37.1	1.0	16.6	1.0	5.7	0.8
Statistical significance ^##^														
Field strength			-		-		-		-		-		-	
Frequency			-		-		-		-		-		-	
Treatment time			-		-		-		-		-		-	

^#^: These treatments led to a partially burnt product which was not acceptable; ^##^: n.s. (non-significant); n.d.: not determined.

## Data Availability

The data presented in this study are available on request from the corresponding author.

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
