# Peer review of "Investigation of Pulsed Electric Field Conditions at Low Field Strength for the Tenderisation of Beef Topside"

_foods, 2022, doi:10.3390/foods11182803_

Round 1

Reviewer 1 Report

The authors evaluated "Investigation of pulsed electric field conditions at low-field strength for the tenderisation of beef topside". The topic is interesting and time relevant, because despite some similar studies have been carried out over the last 5 years, there is still a need to elucidate if PEF can be an interesting tool to improve meat tenderisation, especially at an industrial scale.

Based on the interesting work, I only can tell, the authors did a great work and Introduction, M&M, R&D and Conclusions sections are overall of a great quality.

The only missing points are:

-Why did the authors selected three low field strengths (0.25, 0.50 and 1.00 kV/cm) and did not use higher electric fields (2-3 kV/cm). Not clearly shown the initial temperature and energy inputs reached. 

-For future studies, I am aware about the great experimental work the study involves but it would be advisable to include more analyses to evaluate nutritional and quality characteristics, especially thinking in scaling-up the process.

Author Response

General Comments:

We sincerely appreciate the reviewers' and the editors' efforts in reviewing this manuscript and all the helpful suggestions to improve the quality of this manuscript. We have now responded to all the comments and corresponding revisions have also been made in the manuscript. These revisions are marked in red fonts in the revised manuscript. We are grateful for the speedy and high-quality work done by the editorial team. Thank you.

Reviewer 1

The authors evaluated "Investigation of pulsed electric field conditions at low-field strength for the tenderisation of beef topside". The topic is interesting and time relevant, because despite some similar studies have been carried out over the last 5 years, there is still a need to elucidate if PEF can be an interesting tool to improve meat tenderisation, especially at an industrial scale.

Based on the interesting work, I only can tell, the authors did a great work and Introduction, M&M, R&D and Conclusions sections are overall of a great quality.

Response: The authors would like to thank the reviewer for his/her positive comments on our manuscript. We also believe that more R&D efforts are required to establish the use of PEF for meat tenderisation, and thus, a promising R&D area with a direct potential application. Per your comments, we have made some modifications to the article. Here follows a point-to-point reply to your queries.

The only missing points are:

-Why did the authors selected three low field strengths (0.25, 0.50 and 1.00 kV/cm) and did not use higher electric fields (2-3 kV/cm). Not clearly shown the initial temperature and energy inputs reached. 

Response: We want to remark that the temperature increases for the different treatments (with initial product temperature at around 4°C) are indeed reported in Table 2 and the energy inputs in Table 1.

Concerning the use of low field strength, we decided to use low field strengths as a way of affecting meat texture at the start, based on data from literature and our own experimental data. In addition, using higher electric fields while keeping the treatment time will result in higher energy inputs, potentially leading to meat burning (as shown in Table 2). In fact, the treatment applied at the highest voltage (1 kV) and time (50 ms) led to a partially burnt product which was not acceptable (this fact is indicated as a footnote in Tables 2-7).

Of course, higher energy inputs with shorter processing times could have been applied, but that will require further investigation.

-For future studies, I am aware about the great experimental work the study involves but it would be advisable to include more analyses to evaluate nutritional and quality characteristics, especially thinking in scaling-up the process.

Response: We sincerely appreciate the reviewer's comments, and we will consider their advice in future work as nutritional characteristics are paramount for defining food quality.

Reviewer 2 Report

Article foods-1852052- Investigation of pulsed electric field conditions at low-field strength for the tenderisation of beef topside

The authors propose to study the effect of different PEF treatment conditions (field strengths, frequencies, and treatment times) on the tenderisation and other meat quality traits.

In my opinion, the manuscript is very well written, the material and methods are adequate to achieve the objectives proposed, the results are presented appropriately and the discussion is adequate. My main concern relates to the origin/characterization of the samples collected (see comments).

I am not a native English speaker. Having that in consideration, I consider that the writing is good and adequate for an international journal. So, I found the article interesting and the subject worthy to be published after minor revision.

Comments:

Abstract: the authors study the effect of different PEF conditions mainly in tenderness but in the abstract, the authors do not refer under which PEF treatment the increase in tenderness was observed (only referred to the time of storage), please add this important information, referred in the conclusions section, to the abstract.

Line 51 – ‘between enzymes and substrate’ replace by ‘‘between enzymes and substrates

Line 68-72; 358-359; 365-366; 369 replace ‘M.(in italic) to m. (in italic)

Line 89 – ‘0-2 tooth, females, please instead indicate the age of the animals (and complete the information with animal breed and carcass weight if known).

During the acquisition of topside beef, it is very important that the carcasses have the same time post mortem (if not, the muscles have different aging times and so different tenderisation rates), the authors omit this relevant information, please indicate the time post mortem of samples at the time of sampling.

Line 138 – ‘with pH measurements made per individual sample.’ I think this sentence is missing the number of measurements in each sample.

Line 170 – ‘After cooking, samples were cooled in an ice slurry for 20 min and the weight recorded to determine cook loss’ I think that the samples cooked for shear force measurement are used also for cook loss determination, please mention it.

Line 203-205 – ‘The drip loss (measured as Δweight) was slightly higher in PEF-treated meat, possibly as a result of the electroporation of the muscle cells, especially after PEF treatment at the highest field strength (1 kV/cm) and the longer treatment times of 30 and 50 ms (-0.7 and -0.9 g, and -9.4 and -2.9 g, respectively, Table 2).’ I think that protein denaturation that occurs in the treatments with the highest increase in meat temperature can also be responsible for weight loss and not only electroporation.

Table 2 and others ‘.Treatment conditions’ remove the full stop.

Line 404 – ‘In addition, PEF could promote the release of Ca2+ and μ-calpain early post-mortem’. as it is written, this sentence suggests that PEF releases calpain. The calpains are in the sarcoplasm and not enclosed like the cathepsins. I think it would be more correct to say ‘In addition, PEF could promote the release of Ca2+ and promotes early post-mortem μ-calpain activity

Author Response

General Comments:

We sincerely appreciate the reviewers' and the editors' efforts in reviewing this manuscript and all the helpful suggestions to improve the quality of this manuscript. We have now responded to all the comments and corresponding revisions have also been made in the manuscript. These revisions are marked in red fonts in the revised manuscript. We are grateful for the speedy and high-quality work done by the editorial team. Thank you.

Reviewer 2

Article foods-1852052- Investigation of pulsed electric field conditions at low-field strength for the tenderisation of beef topside

The authors propose to study the effect of different PEF treatment conditions (field strengths, frequencies, and treatment times) on the tenderisation and other meat quality traits.

In my opinion, the manuscript is very well written, the material and methods are adequate to achieve the objectives proposed, the results are presented appropriately and the discussion is adequate. My main concern relates to the origin/characterisation of the samples collected (see comments).

I am not a native English speaker. Having that in consideration, I consider that the writing is good and adequate for an international journal. So, I found the article interesting and the subject worthy to be published after minor revision.

Response: The authors would like to thank the reviewer for their positive comments, suggestions and recommendations on our manuscript. Here follows a point-to-point reply to your queries.

Comments:

Abstract: the authors study the effect of different PEF conditions mainly in tenderness but in the abstract, the authors do not refer under which PEF treatment the increase in tenderness was observed (only referred to the time of storage), please add this important information, referred in the conclusions section, to the abstract.

Response: This question is not simply answered, as many simultaneous interactions play a role, and further explanations to understand the results are required (as presented in the sections results and further explained in the discussion). We have now stated in the abstract that field strength (0.25-0.5 kV/cm) seems to work better, affecting meat structure.

Line 51 – 'between enzymes and substrate' replace by "between enzymes and substrates

Response: Thank you for detecting this edit. The change has been done in the revised manuscript.

Line 68-72; 358-359; 365-366; 369 replace ‘M. (in italic) to m. (in italic)

Response: Thank you for detecting this edit. The change has been done in the revised manuscript.

Line 89 – '0-2 tooth, females, please instead indicate the age of the animals (and complete the information with animal breed and carcass weight if known).

Response: In Australia, animals are often not born in captivity, and the exact birth date is, therefore, unknown. In addition, the grading system in Australia does not consider age as an absolute numeric value. However, it considers age as an indication of an animal's maturation state, which is related to meat quality and can be rated by assigning a tooth category (as reported).

Animal breed (mixed breed) and carcass weight (235 ± 23 kg) plus pH (5.58 ± 0.03) and temperature (5.56 ± 1.01 °C) along with topside weight (7.59 ± 0.44 kg) at harvest were recorded and have now been included in the revised version of the manuscript (lines 89-92).

During the acquisition of topside beef, it is very important that the carcasses have the same time post mortem (if not, the muscles have different aging times and so different tenderisation rates), the authors omit this relevant information, please indicate the time post mortem of samples at the time of sampling.

Response: The post-mortem time was 24-48 hours for all cases (information included in the revised manuscript, line 90) to assure that no meat maturation was taking place and an equivalent time post-mortem for all the muscles included in the study.

Line 138 – 'with pH measurements made per individual sample.' I think this sentence is missing the number of measurements in each sample.

Response: The muscle pH was measured twice in each muscle (line 93).

Line 170 – 'After cooking, samples were cooled in an ice slurry for 20 min and the weight recorded to determine cook loss' I think that the samples cooked for shear force measurement are used also for cook loss determination, please mention it.

Response: The same samples cooked for WB shear force measurement were used for cook loss determination (line 172)

Line 203-205 – 'The drip loss (measured as Δweight) was slightly higher in PEF-treated meat, possibly as a result of the electroporation of the muscle cells, especially after PEF treatment at the highest field strength (1 kV/cm) and the longer treatment times of 30 and 50 ms (-0.7 and -0.9 g, and -9.4 and -2.9 g, respectively, Table 2).' I think that protein denaturation that occurs in the treatments with the highest increase in meat temperature can also be responsible for weight loss and not only electroporation.

Response:  We agree with the reviewer that heat-induced protein denaturation could cause drip loss in overheated samples. A remark has been added to specify this (line 209-210).

Table 2 and others '.Treatment conditions' remove the full stop.

Response: This has now been deleted as advised.

Line 404 – 'In addition, PEF could promote the release of Ca2+ and μ-calpain early post-mortem'. as it is written, this sentence suggests that PEF releases calpain. The calpains are in the sarcoplasm and not enclosed like the cathepsins. I think it would be more correct to say 'In addition, PEF could promote the release of Ca2+ and promotes early post-mortem μ-calpain activity

Response: Thank you for this suggestion. We agree with the reviewer and have rewritten the sentence to distinguish between Ca2+ release and activation of calpains (line 409).